# Date (*Phoenix dactylifera* L. cv. Medjool) Seed Flour, a Potential Ingredient for the Food Industry: Effect of Particle Size on Its Chemical, Technological, and Functional Properties

**DOI:** 10.3390/plants13030335

**Published:** 2024-01-23

**Authors:** Nuria Muñoz-Tebar, Laura Candela-Salvador, José Ángel Pérez-Álvarez, José Manuel Lorenzo, Juana Fernández-López, Manuel Viuda-Martos

**Affiliations:** 1IPOA Research Group, Agro-Food Technology Department, Instituto de Investigación e Innovación Agroalimentaria y Agroambiental (CIAGRO-UMH), Miguel Hernández University, Orihuela, 03312 Alicante, Spain; laura.candela03@goumh.umh.es (L.C.-S.); ja.perez@umh.es (J.Á.P.-Á.); j.fernandez@umh.es (J.F.-L.); 2Centro Tecnológico de la Carne de Galicia, Avd. Galicia No. 4, Parque Tecnológico de Galicia, San Cibrao das Viñas, 32900 Ourense, Spain; jmlorenzo@ceteca.net; 3Área de Tecnología de los Alimentos, Facultad de Ciencias de Ourense, Universidade de Vigo, 32004 Ourense, Spain

**Keywords:** date seed flour, polyphenolic profile, mineral profile, antioxidant activity, particle size, functional properties

## Abstract

The objective of this work was to evaluate the effect of particle size on the chemical composition, fatty acid and polyphenol profile, physicochemical and techno-functional properties, and antioxidant capacity of flour obtained from date seeds. The date seed flours obtained had a high content of total dietary fiber (67.89–76.67 g/100 g), and the reduction in particle size decreased the moisture and protein contents, while the fat, mineral (Ca, Fe, Zn, and Mg), and fatty acid contents were significantly increased, with oleic acid being the highest. Water activity increased with decreasing particle size, and the finest flour (<210 mm) tended to be yellowish and reddish. The water- and oil-holding capacities decreased in the flours with the smallest particle size compared to the largest sizes. The main polyphenolic compounds in all the samples were catechin, epicatechin, and epigallocatechin-3-gallate. The antioxidant activity significantly improved with reductions in the particle size of the date seed flour, with the ABTS, DPPH, and FRAP values ranging between 8.99 and 20.68, 0.66 and 2.35, and 1.94 and 4.91 mg Trolox equivalent/g of date seed flour. The results of the present study suggest that the flour obtained from date seeds cv. Medjool can be a valuable co-product for the food industry due to its fiber content, essential fatty acids, and bioactive compounds that can help reduce the amount of waste generated, promoting the circular economy in the food chain.

## 1. Introduction

Eco-efficiency is a business management philosophy based on offering short- and medium-term solutions useful for innovation within a transition process towards sustainable development. Its main strategic objective is to achieve higher performance with less use of natural resources and less waste production throughout a given process [1]. The need to improve the sustainability of agriculture and eco-efficiency in food production has led to the search for the valorization of several co-products generated in industrial processes [2]. Currently, food industries are undergoing constant changes in order to adapt to the needs of consumers, who demand “more natural” foods, as well as to the need for proper waste management that can help preserve the environment. Terms such as sustainability are becoming commonplace in the agri-food sector, industries, governments, and the scientific community, which are making great efforts to minimize the impact on the environment by mainly focusing on issues such as innovation, biodiversity, and the local economy. Globalization, along with emerging consumption trends, has caused the food industry to increase the use of raw materials or primary products, leading to an increase in co-products and problems with their management and utilization. Initiatives to valorize these co-products and give them a new value open new lines of work while creating new business opportunities for the food industry [3].

The date fruit of the date palm (*Phoenix dactylifera* L.) is a product of great cultural, social, and economic importance in many countries, having special relevance in arid regions. The rest of the world production (10%) is the object of a very active international trade, basically supported by the presence of Muslim populations in areas where date production is not feasible [4]. There is also a traditional European market for date consumption mainly as dried fruit. In Spain, these two markets are combined in the Levante region, where dates from the Elche palm grove are traditionally consumed fresh. Date cultivation is a source of income for the local population and represents an economic alternative of great potential in a region where agriculture requires viable options. Export markets are essentially supplied by dried and semi-soft dates, which are easier to preserve than fresh dates due to their high water content [5]. The commercialization of fresh dates, which has a very important competitive advantage, is nonetheless associated with a high percentage of waste (in the 2018–2019 season, it was estimated at 40%) that is generated due to not meeting the high-quality standards established for this trade. It is, therefore, expected to have, in a short period, a large quantity of dates that are not qualified for marketing as fresh but completely suitable for processing. In this sense, the industrialization of dates will generate a large quantity of another co-product, namely, date seeds. Date seeds represent 10% of the weight of the fruit. They are rich in oil, proteins, minerals, dietary fiber, and bioactive compounds, which means that they could be reused in food processing and manufacturing processes to develop novel foods with health benefits [6]. One of the main reasons why date seeds have a high nutritional value is their dietary fiber content, which makes them suitable for the developments of fiber-enriched or dietetic foods [7,8]. Another important aspect is their bioactive compound content of mainly mono- and polyunsaturated fatty acids and polyphenolic compounds [9,10,11]. The composition of oil extracted from the seeds can vary due to the different crop varieties since some of them contain greater amounts of unsaturated than saturated fatty acids. Overall, oleic and linoleic acids stand out as the major unsaturated fatty acids [9]. With regards to their content of phenolic compounds and antioxidant capacity, date seeds contain a high amount of phenols, including gallic acid, protocatechuic acid, *p*-hydroxybenzoic acid, caffeic acid, *p*-coumaric acid, and ferulic acid among others [11]. However, it should be borne in mind that the composition of phenolic compounds in date seeds can vary widely due to several factors such as the origin, variety, ripening stage, processing, and the experimental analytical conditions used [12]. Due to the high content of bioactive compounds, this flour could be used as a potential ingredient in the development of functional food mainly in the meat and bakery industries to improve the shelf life of a product and provide potentially beneficial compounds for health.

After conducting a thorough literature search, it was noticed that the literature concerning the composition of date seed flour is still very limited, especially regarding the effect of particle size on its properties since no studies have been found so far. Therefore, the objective of the present work was to evaluate the effect of particle size reduction on the chemical composition, physicochemical, techno-functional, and antioxidant properties of date seed flour cv. Medjool obtained from the Elche palm grove.

## 2. Results

### 2.1. Chemical Composition

The effect of particle size on the chemical composition of date seed flour is shown in Table 1. The moisture content of flours and powders is one of the most critical properties since it can impair the chemical and physical properties of the foods in which they are incorporated. It is also one of the critical factors for their shelf life and stability [13]. In the samples analyzed, the moisture content was significantly affected by particle size reduction (*p* < 0.05) ranging from 12.03 to 9.06 g/100 g. The flours with the smallest particle size (MLDSF and LDSF) were the only ones whose values were within the moisture limits for safe storage (8.55–10.18%), as stated by Nabil et al. [13]. This reduction in moisture content as particle size decreases agrees with the results reported by Lucas-Gonzalez et al. [14] for persimmon flour. This reduction may be due to a reduction in the specific surface area.

The protein content (Table 1) showed the same behavior noticed for the moisture (Table 1), with a significant reduction (from 4.47% to 3.67%) as the particle size was reduced. However, the fat content (Table 1) showed an opposite behavior, with a significant increase (*p* < 0.05) with a reduction in the particle size, exhibiting the highest value in the date seed flour with the finest particle size (Ø < 0.21 mm). This fact was also observed by Botella-Martínez et al. [15] in flours obtained from cocoa bean shells with particle sizes ranging between 0.70 and 0.417 mm. For the ash content (Table 1), no statistical differences (*p* > 0.05) were found between any of the particle sizes analyzed. The moisture and ash contents were higher compared to those determined in other studies using dates of the same variety, although the protein content was similar to the values contrary to fat that was much lower with values up to 2 times smaller (5.01% vs. 10.19%) than those found in the literature [16,17]. Concerning the total dietary fiber (Table 1), a significant reduction was observed with a decrease in the particle size up to <0.21 mm compared to the highest one (76.67 vs. 67.89 g/100 g), while a medium particle size of date seed flours (MHDSF and MLDSF) showed similar values with no statistical differences (*p* > 0.05) between them. Comparing the values obtained with those found in the literature, the dietary fiber content for all particle sizes analyzed was higher than reported by Vinita and Punia [18] and Salomón-Torres et al. [16] in date seed flours from the same cultivar as the one used for the present work with values of 66.46 and 65.46 g/100 g, respectively. As it is widely known, dietary fibers play a key role as an ingredient in the formulation of new functional foods because of their technological properties and health benefits such as their positive influence on cardiovascular disease, improvement of intestinal health and insulin response, influence on fat and cholesterol absorption, and reduction in the risk of developing some types of cancer [19]. Thus, the high total dietary fiber content of date seed flour indicates that it could be a valuable ingredient in the formulation of novel fiber-enriched foods. 

#### 2.1.1. Mineral Profile

Table 2 shows the effect of particle size reduction upon the mineral content of date seed flours cv. Medjool cultivated on the Elche palm grove. For sodium, manganese, copper, and zinc, no statistical differences (*p* > 0.05) were observed between the different particle sizes analyzed. However, for calcium, iron, and potassium, as the particle size decreased, the concentration of these three compounds increased, showing statistically significant differences (*p* < 0.05) between the different particle sizes. Magnesium showed the highest value (*p* < 0.05) for the lowest particle size (LDSF) with no differences (*p* > 0.05) between the other sizes.

For all the samples analyzed, potassium showed the highest (*p* < 0.05) values, followed by sodium and calcium, and at the other extreme were zinc and copper, which showed the lowest (*p* < 0.05) values without statistical differences (*p* > 0.05) between them. After searching the literature, no studies on the effect of particle size reduction on the mineral composition of date seed flour have been found so far. However, Bijami et al. [20] conducted a study of the mineral composition of date seed at four development stages (hababouk, kimri, khalal, and tamar), in which they reported higher values of calcium, zinc, and magnesium but lower values of iron, potassium, copper, and manganese than the values obtained in the present study. Similarly, Vinita and Punia [18] and Salomón-Torres et al. [16] analyzed the mineral content of date seed cv Medjool and obtained lower values of calcium, iron, potassium, manganese, and sodium but higher values of zinc, copper, and magnesium.

#### 2.1.2. Fatty Acid Profile

The results of the analysis of the fatty acid content of date seed flour influenced by particle size are presented in Table 3 and showed a significant increase (*p* < 0.05) in all the fatty acids when the particle size was reduced from 1.16 mm to <0.21 mm.

The most abundant fatty acids in all the flours were oleic acid (C18:1) and lauric acid (C12:0), increasing from 11.23 to 22.70 mg/g date seed flour and from 5.48 to 10.88 mg/g date seed flour, respectively. Likewise, stearic acid (C18:0) was the one found in the lowest proportion ranging from 0.92 mg/g to 1.75 mg/g date seed flour. Therefore, the results show that date seed flour is characterized by a higher percentage of monounsaturated fatty acids (oleic acid (C18:1 n-9) followed by saturated fatty acids (lauric acid, myristic acid, and palmitic acid) and polyunsaturated fatty acids consisting only of linoleic acid (C18:3 n-3). The use of oils with high oleic acid contents for food formulations has increased due to their high oxidative stability and nutritional importance since they are characterized as being one of the most important unsaturated fatty acids due to their health benefits such as their potential to reduce blood cholesterol [21].

Regarding the effect of particle size reduction on the fatty acid content, significant increases for these compounds were previously reported in flours obtained from cocoa shell co-products and chia seeds [15,22] (supporting the fact that particle size plays a key role in the efficiency and yield of fatty acid extraction. These results also prove that it would be more attractive to use flours with smaller particle sizes with the aim to develop novel foods enriched with essential fatty acids, such as the one belonging to the omega-3 group. The fatty acid profile of the date seed flour used in the present study was similar to the fatty acid composition determined by Salomón-Torres et al. [17] in Medjool date seeds, although their values were higher compared to our values.

### 2.2. Physicochemical Properties

Table 4 shows the results of the effect that the particle size reduction exerted on the physicochemical properties of the flour obtained from date seeds. As it can be observed, pH values ranged from 4.93 to 5.55 with no statistical differences between the flours, with the particle sizes oscillating from 0.70 mm to <0.21 mm (MHDSF, MLDSF, and LDSF) and HDSF being the one with the highest pH value (*p* < 0.05). The values obtained in the present study were comparable to the ones obtained for cocoa shell flour at different particle sizes (>0.70 mm to <0.417 mm) [15].

For the water activity, the decrease in the particle size resulted in a significant (*p* < 0.05) increase, and the LDSF was the one with the highest value (0.555 vs. 0.523). Increased water activity values due to particle size reduction have been previously reported by Botella-Martínez et al. [15] in flours obtained from cocoa shell co-products at different particle sizes.

In relation to the color coordinates, the lightness (L*) values were similar for all particle sizes while the a* and b* coordinates increased significantly when the particle size of the date seed flour was reduced (Table 4). In the case of coordinate a*, it was observed that the highest values occurred in the MLDSF and LDSF flours (0.42 mm and <0.21 mm), while the HDSF and MHDSF flours had the lowest values (8.53 and 9.09, respectively). However, the flour with the largest particle size (HDSF: 1.16 mm < Ø > 0.70 mm) had the lowest values for coordinate b* (*p* < 0.05), whilst the values of the rest of the sizes were similar (13.09–13.59). These results indicate that date seed flours with smaller particle sizes tend to be more yellowish and reddish than those with larger sizes. In this sense, other studies have reported increases in the coordinates a* and b* in persimmon flour and unripened banana flour when their particle size was reduced [14,23]. This fact could be related to the release of pigments that may occur during the milling and sieving process, as previous studies have reported that the milling process has a strong influence on several flour components such as proteins or pigments that in turn could modify their color properties [15,24]. 

Regarding the values reported for date seeds, the water activity and lightness were higher than those found by Bouaziz et al. [25] and Salomón-Torres et al. [16], while pH, yellowness and redness values were lower compared to the values obtained by other authors for Deglet Nour date seeds [25].

### 2.3. Techno-Functional Properties

The effect of the particle size on the techno-functional properties of date seed flour is shown in Figure 1 as the measurement of the water-holding capacity (WHC; Figure 1A), oil-holding capacity (OHC, Figure 1B), and swelling capacity (SWC, Figure 1C). 

A significant reduction (*p* < 0.05) in WHC and OHC properties was obtained for the finest date seed flour (LDSF) with similar values for the other particle sizes (HDSF, MHDSF, and MLDSF), while SWC was increased. In the case of WHC (Figure 1A), the values ranged between 2.25 and 2.54 g water/g date seed flour, and the lower value found for the finest size may be related to its lower percentage of fiber compared to the rest of the flours. The relatively high WHC values obtained support the fact that date seed flour could be used as an ingredient in food formulations to improve their sensory properties. The OHC (Figure 1B) and SWC (Figure 1C) values oscillated from 1.40 to 1.64 g oil/g and from 22.61 to 25.12 mL/g date seed flour, respectively. 

A reduction in the WHC and OHC capacity has been previously observed by Bouaziz et al. [26] for defatted date seed flour, Botella-Martínez et al. [15] for cocoa bean shell flour, and by Lucas-González et al. [14] for persimmon flour, and according to other studies, the reduction in these properties may be caused by several factors such as (i) a reduction in direct absorption; (ii) a reduction in capillary forces; and (iii) a reduction in particle surface properties [27]. When comparing the techno-functional properties of the flour obtained from Medjool dates, it was noticed that the water- and oil-holding capacities as well as the swelling capacity were better than the values reported by Kelany and Yemiş [28] for date seed cv. Saidy but lower in comparison with the values obtained for the date seed Deglet Nour variety [25]. However, Gökşen et al. [29] reported that the WHC of date seed flours obtained from cultivars Safawi, Suhgai, and Mebruum were higher than those obtained in this study, with values ranging between 5.96 and 6.87 g/g date flour.

### 2.4. Polyphenolic Profile

Table 5 shows the effect of particle size on the polyphenolic profile of date seed flour cv. Medjool cultivated on the Elche palm grove. Except for 3-*O*-caffeoylshikimic acid, quercetin 3-β-d-glucoside, and quercetin 3-rhamnoside, an increase in the extraction amount of polyphenolic compounds was achieved with the reduction in particle size. i.e., the highest concentrations for these compounds were found in samples with the smallest particle size. This fact has been reported by several authors such as Zaiter et al. [30] for *Hedera helix* and *Scrophularia nodosa* powder and Pătrăuţanu et al. [31] for spruce bark. The extraction efficiency can be significantly influenced by particle size, which plays a crucial role in governing mass transfer kinetics. Additionally, particle size determines the accessibility of solvent, enhancing permeability and diffusivity for bioactive compounds [32].

In all the samples analyzed, thirteen compounds were identified that can be classified as one dihydroxybenzoic acid (protocatechuic acid), two hydroxycinnamic acids (caffeic and *p*-coumaric acids), four flavan-3-ols (gallocatechin-3-gallate, epigallocatechin-3-gallate, catechin, and epicatechin), three flavonols (quercetin 3-rutinoside, quercetin 3-β-d-glucoside, and quercetin 3-rhamnoside), and three compounds derived from caffeic acid and shikimic acid (3-*O*-caffeoylshikimic acid, 4-*O*-caffeoylshikimic acid, and 5-*O*-caffeoylshikimic acid).

The principal polyphenolic compound found (*p* < 0.05) in all the date seed flour samples examined was catechin followed by epigallocatechin-3-gallate and epicatechin. In the scientific literature, there was much variability in the polyphenolic profile of date seeds. Thus, Ranasinghe et al. [33] reported that the main polyphenolic compounds found in the defatted date seed flour cultivars Khalas, Fardh, and Khenaizi and extracted using ultrasonication were benzoic acid (23.01–30.96 mg/100 g) and catechin (9.39–25.62 mg/100 g). In a similar study, Bouhlali et al. [34] found that the main polyphenolic compounds present in the date seeds cultivars *Boufgous, *Bousthammi*, *Jihl*,* and *Medjool* were p-coumaric acid (116.96–143.60 mg/100 g), caffeic acid (59.40–88.64 mg/100 g), and gallic acid (10.35–17.62 mg/100 g). Al Juhaimi et al. [35] mentioned that the hydroxycinnamic acids gallic acid and syringic acid were the principal compounds of date seed extracts from eleven different cultivars cultivated in Algeria, Libya, Morocco, Pakistan, and Sudan.

The variability in the polyphenolic profiles of date seeds could be explained due to several factors such as the cultivar, growing conditions, water availability, diseases, maturity, harvesting times, storage conditions, geographic origin, light, temperature, fertilizer, soil type, and the extraction system and method of analysis [36].

### 2.5. Antioxidant Activity

Particle size reduction had a positive effect on antioxidant activity, with a significant increase (*p* < 0.05) in the results obtained with ABTS (Figure 2A), DPPH (Figure 2B), and FRAP assays (Figure 2C). This fact has been observed with other flours where the particle size has been analyzed. Thus, an improvement in the antioxidant capacity by reducing the particle size has also been noticed by Savlak et al. [23] for unripe banana flour and by Lucas-González et al. (2018) [14] for persimmon flour. 

The highest antioxidant capacity values were obtained for the finest particle size, reaching values of 20.68 mg Trolox equivalent/g for ABTS, 2.35 Trolox equivalent/g for DPPH, and 4.91 Trolox equivalent/g date seed flour in the case of the FRAP assay. This behavior may be due to the fact that the matrix is less altered at larger particle sizes, thereby lowering the release of bioactive compounds during the extraction. As occurred for the polyphenolic profile, there was a great variability in the antioxidant properties of extracts obtained from date seeds. In this sense, Djaoudene et al. [37] carried out a study to determine the antioxidant capacity of extracts obtained from date seed (cv Ourous, Ouaouchet, and Oukasaba) cultivated in Argelia by means of ABTS and DPPH assays. These authors reported values, for the ABTS assay, ranging between 0.56 g and 066 G TE/g, while for the DPPH assay, the values reported varied between 0.57 and 0.81 g TE/g. Similarly, Bouhlali et al. [38] reported that the antioxidant capacity, measured with ABTS and FRAP assays, of date seed extracts obtained from varieties Bousthammi and Medjool ranged between 5.59 and 8.02 mmol TE/100 g for the ABTS assay, whilst for the FRAP assay, the values achieved varied between 10.96 and 22.86 mmol TE/100 g. In both studies, the values obtained were higher than those obtained in this study.

Based on the results, it could be stated that particle size has a critical role in improving and increasing the extraction efficiency of bioactive compounds and thus their antioxidant potential, with the smaller sizes being a better option as a functional ingredient for the development of foods with antioxidant properties.

## 3. Materials and Methods

### 3.1. Date Seed Flour

The dates (cv. Medjool) without adequate commercial size, free from damage and infections, were obtained from the Elche palm groove. The seeds were manually separated from the pulp and dried for 48 h at 55 °C. Subsequently, they were ground and passed through different sieves to obtain four date seed flours with diverse particle sizes: HDSF, highest date seed flour (1160 mm < Ø > 0.70 mm); MHDSF, medium-high date seed flour (0.70 mm < Ø > 0.42 mm); MLDSF, medium-low date seed flour (0.42 mm < Ø > 0.21 mm); and LDSF, lowest date seed flour (Ø < 0.21 mm). Finally, the date seed flours were stored in vacuum packaging until their analysis. Figure 3 shows the different date seed flours obtained.

### 3.2. Chemical Composition 

The chemical composition (moisture, protein, fat, ash, and total dietary fiber content) of the date seed flours with different particle sizes was analyzed in triplicate following the AOAC methods [39]. For the mineral composition, samples were digested by microwave with 67% nitric acid and 33% hydrogen peroxide followed by quantification using inductively coupled plasma mass spectrometry (ICP-MS) with a Shimadzu MS-2030 (Shimadzu, Kyoto, Japan) according to the following operating conditions: carrier gas 0.70 L/min; plasma gas 9.0 L/min; auxiliary gas 1.10 L/min; radio frequency 1.2 kW; and energy filter 7.0 V. The measurements were performed in triplicate, and the mineral content was expressed as mg/100 g of date seed flour. 

For the analysis of the fatty acid profile, the lipids were extracted according to Folch et al. [40] and methylated following the procedure described by the AOAC method 969.33 [41]. Then, the fatty acid methyl esters (FAMEs) were quantified by gas chromatography (GC) with the same conditions described by Pellegrini et al. [42]. The measurements were carried out in triplicate, and the results were expressed as g FAME/100 g fat. 

### 3.3. Physicochemical Properties

The pH of the date seed flours with different particle sizes was determined in triplicate by using a GLP21 (Crison, Barcelona, Spain) pH meter with the samples diluted 1:10 in deionized water. The color (CIEL*a*b* coordinates) of the date seed flour was measured in triplicate with a CM-700 Minolta spectro-photocolorimeter (Minolta, Osaka, Japan) equipped with an illuminant D_65_ and an observation angle of 10°. Lastly, the water activity (aw) of the date seed flours with different particle sizes was determined in triplicate with a Novasina TH-500 (Novasina, Axair Ltd., Pfaeffikon, Switzerland) water activity analyzer at room temperature. 

### 3.4. Techno-Functional Properties

The water-holding capacity (WHC), oil-holding capacity (OHC), and swelling capacity (SWC) of the date seed flours with different particle sizes were evaluated in triplicate using the same method described by Muñoz-Bas et al. [43]. The results were expressed as the weight of water held (WHC) or oil held (OHC) by 1 g of date seed flour and as mL per gram of date seed flour in the case of SWC. 

### 3.5. Polyphenol Content

Polyphenolic compounds were extracted with the method described by Genskowsky et al. [44] with slight modifications. Briefly, 2 g of the date seed flour was mixed with 20 mL of methanol:water (80:20) and stirred overnight. After this time, the samples were centrifugated at 8000 rpm/10 min/4 °C, and the supernatant was recovered. Then, the pellet was resuspended with acetone:water (70:30), and the samples were homogenized at 20,000 rpm for 2 min in a homogenizer (Ultra-Turrax T25 BASIC, IKA-Werke GmbH & Co. KG, Staufen, Germany). Finally, the samples were centrifuged again with the same conditions to recover the supernatant, which was mixed with the previous one. Subsequently, the samples were passed through a C-18 Sep-Pak cartridge to avoid potential interferences during chromatographic analyses. Quantification of the polyphenolic compounds was carried out in triplicate on an Agilent HPLC 1200 series chromatograph (Agilent Technologies, Santa Clara, CA, USA) using the same conditions described by Genskowsky et al. [44].

### 3.6. Antioxidant Activity

The antioxidant properties of the date seed fours were assessed in triplicate using three different antioxidant assays (DPPH, ABTS, and FRAP). The DPPH and ABTS assays were performed following the methods proposed by Brand-Williams et al. [45] and Gullon et al. [46], respectively, and the FRAP assay was evaluated using the same method described by Oyaizu [47]. All the results were expressed as mg Trolox equivalents (TE)/g date flour.

### 3.7. Statistical Analysis

Statistical analysis of the data was performed using SPSS (IBM SPSS Statistics version 26). ANOVA (one way) was calculated using a confidence level of 95% to determine any significant differences (*p* < 0.05), and the Tukey test was carried out to determine the differences between the different particle sizes (HDSF: 1.16 mm < Ø > 0.70 mm; MHDSF: 0.70 mm < Ø > 0.42 mm; MLDSF: 0.42 mm < Ø > 0.21 mm; and LDSF: Ø < 0.21 mm) of date seed flour. 

## 4. Conclusions

The results of the present study demonstrate that date seed flour is an emerging ingredient with potential application in the development of new foods due to its high dietary fiber content, essential fatty acid composition, and antioxidant capacity. However, depending on its intended application, it is necessary to bear in mind the effect that the particle size reduction caused on the technological and functional properties of the flour obtained from the date seeds. Overall, it could be concluded that date seed flour at a finer particle size could be a more suitable option for incorporation as an ingredient in the development of healthier meat products as well as in bakery products, although further studies should be carried out to verify this hypothesis.

## Figures and Tables

**Figure 1 plants-13-00335-f001:**
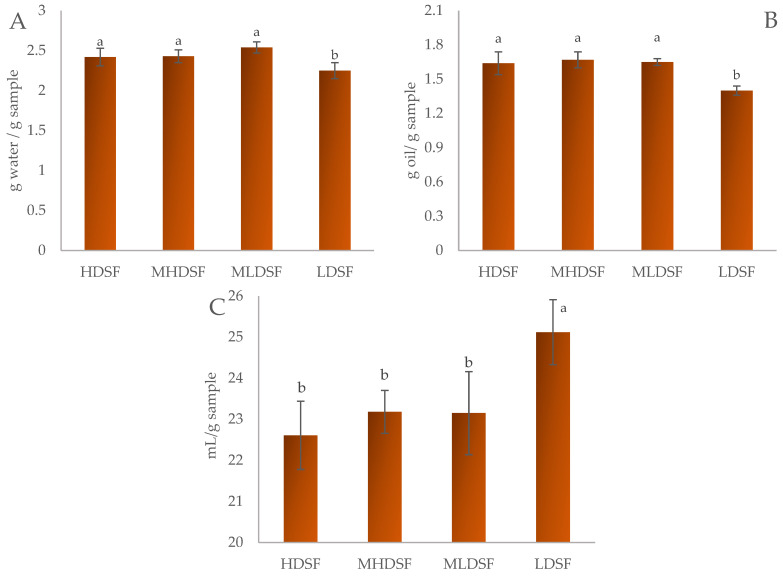
Effect of particle size on the techno-functional ((**A**): water-holding capacity; (**B**): oil-holding capacity; and (**C**): swelling capacity) properties of date seed flour cv. Medjool cultivated on the Elche palm grove. HDSF: highest date seed flour (1.16 mm < Ø > 0.70 mm); MHDSF: medium-high date seed flour (0.70 mm < Ø > 0.42 mm); MLDSF: medium-low date seed flour (0.42 mm < Ø > 0.21 mm); and LDSF: lowest date seed flour (Ø < 0.21 mm). For each techno-functional property, histograms with different superscripts indicate significant differences (*p* < 0.05) according to Tukey’s multiple range test.

**Figure 2 plants-13-00335-f002:**
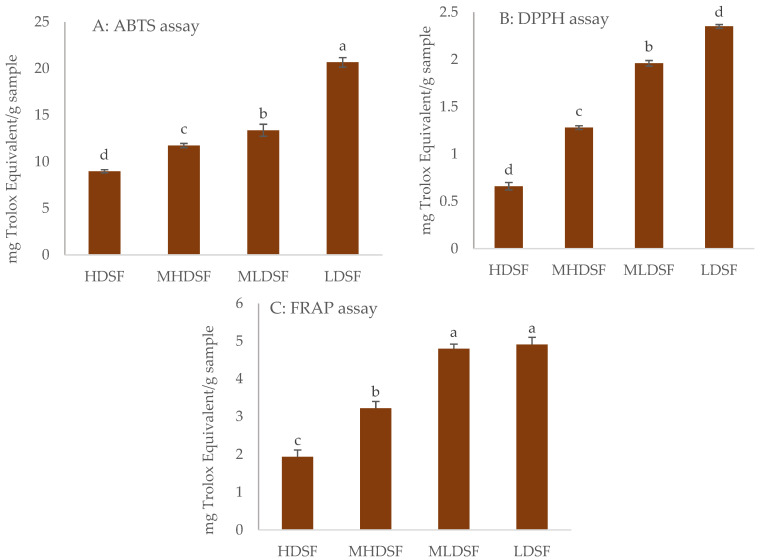
Effect of particle size on antioxidant properties ((**A**): ABTS assay; (**B**): DPPH assay; (**C**): FRAP assay) of date seed flour cv. Medjool cultivated on the Elche palm grove. HDSF: highest date seed flour (1.16 mm < Ø > 0.70 mm); MHDSF: medium-high date seed flour (0.70 mm < Ø > 0.42 mm); MLDSF: medium-low date seed flour (0.42 mm < Ø > 0.21 mm); and LDSF: lowest date seed flour (Ø < 0.21 mm). For each antioxidant assay, histograms with different superscripts indicate significant differences (*p* < 0.05) according to Tukey’s multiple range test.

**Figure 3 plants-13-00335-f003:**
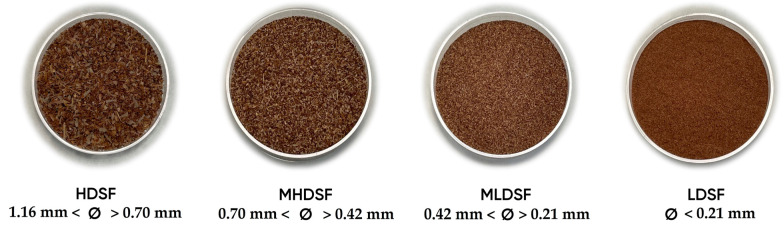
Date (*Phoenix dactylifera* L. cv. Medjool) seed flours with different particle sizes.

**Table 1 plants-13-00335-t001:** Effect of particle size on the chemical composition of date seed flour cv. Medjool cultivated on the Elche palm grove.

	Moisture	Protein	Fat	Ash	Total Dietary Fiber
HDSF	12.03 ± 0.36 ^a^	4.47 ± 0.09 ^a^	2.73 ± 0.27 ^d^	2.87 ± 0.18 ^a^	76.67 ± 0.81 ^a^
MHDSF	10.25 ± 0.11 ^b^	4.01 ± 0.04 ^b^	3.35 ± 0.22 ^c^	2.70 ± 0.12 ^a^	72.78 ± 0.98 ^b^
MLDSF	9.30 ± 0.08 ^c^	3.82 ± 0.05 ^c^	4.19 ± 0.19 ^b^	2.87 ± 0.07 ^a^	70.42 ± 0.75 ^b^
LDSF	9.06 ± 0.12 ^d^	3.64 ± 0.03 ^d^	5.01 ± 0.12 ^a^	2.93 ± 0.48 ^a^	67.89 ± 0.51 ^c^

Values expressed as g/100 g of date seed flour. HDSF: highest date seed flour (1.16 mm < Ø > 0.70 mm); MHDSF: medium-high date seed flour (0. 70 mm < Ø > 0.42 mm); MLDSF: medium-low date seed flour (0.42 mm < Ø > 0.21 mm); and LDSF: lowest date seed flour (Ø < 0.21 mm). ^a–d^ According to Tukey’s multiple range test, values with different superscripts within the same column indicate significant differences (*p* < 0.05).

**Table 2 plants-13-00335-t002:** Effect of particle size on the mineral content of date seed flour cv. Medjool cultivated on Elche palm grove.

	Calcium	Iron	Potasium	Copper	Zinc	Manganese	Magnesium	Sodium
HDSF	157.97 ± 3.55 ^aC^	9.33 ± 0.09 ^aE^	552.13 ± 6.87 ^dA^	0.49 ± 0.05 ^aG^	0.52 ± 0.01 ^aG^	51.98 ± 1.07 ^aD^	8.53 ± 0.37 ^aF^	212.36 ± 13.11 ^aB^
MHDSF	152.56 ± 4.69 ^bC^	10.65 ± 0.43 ^cE^	595.30 ± 5.97 ^cA^	0.53 ± 0.00 ^aG^	0.53 ± 0.01 ^aG^	53.48 ± 1.12 ^aD^	9.09 ± 0.17 ^aF^	192.74 ± 22.71 ^aB^
MLDSF	165.48 ± 5.34 ^cC^	12.74 ± 0.55 ^bE^	623.23 ± 8.84 ^bA^	0.51 ± 0.01 ^aG^	0.54 ± 0.01 ^aG^	53.27 ± 1.26 ^aD^	9.11 ± 0.13 ^aF^	213.70 ± 10.71 ^aB^
LDSF	175.06 ± 2.36 ^dC^	17.63 ± 1.46 ^aE^	642.05 ± 7.90 ^aA^	0.51 ± 0.03 ^aG^	0.55 ± 0.01 ^aG^	52.12 ± 0.23 ^aD^	12.15 ± 0.07 ^bF^	218.98 ± 18.08 ^aB^

Values expressed as mg/100 g of date seed flour. HDSF: highest date seed flour (1.16 mm < Ø > 0.70 mm); MHDSF: medium-high date seed flour (0.70 mm < Ø > 0.42 mm); MLDSF: medium-low date seed flour (0.42 mm < Ø > 0.21 mm); and LDSF: lowest date seed flour (Ø < 0.21 mm). Values with a different lowercase letter within the same column indicate significant differences (*p* < 0.05) according to Tukey’s multiple range test. Values followed with a different capital letter in the same row indicate significant differences (*p* < 0.05) according to Tukey’s multiple range test.

**Table 3 plants-13-00335-t003:** Effect of particle size on the fatty acid profile of date seed flour cv. Medjool cultivated on the Elche palm grove.

	Lauric Acid	Myristic Acid	Palmitic Acid	Stearic Acid	Oleic Acid	Linoleic Acid
HDSF	5.48 ± 0.37 ^aB^	2.57 ± 0.23 ^aC^	2.41 ± 0.15 ^aC^	0.92 ± 0.15 ^aE^	11.23 ± 0.78 ^aA^	2.18 ± 0.12 ^aD^
MHDSF	7.23 ± 0.14 ^bB^	3.49 ± 0.06 ^bC^	3.17 ± 0.11 ^bC^	1.20 ± 0.08 ^bE^	14.91 ± 0.12 ^bA^	2.78 ± 0.18 ^bD^
MLDSF	9.71 ± 0.10 ^cB^	4.62 ± 0.11 ^cC^	4.04 ± 0.12 ^cD^	1.51 ± 0.10 ^cF^	19.07 ± 0.14 ^cA^	3.49 ± 0.09 ^dE^
LDSF	10.88 ± 0.16 ^dB^	5.29 ± 0.04 ^dC^	4.72 ± 0.07 ^dD^	1.75 ± 0.05 ^dF^	22.79 ± 0.23 ^dA^	4.14 ± 0.03 ^dE^

Values expressed as mg/100 g of date seed flour. HDSF: highest date seed flour (1.16 mm < Ø > 0.70 mm); MHDSF: medium-high date seed flour (0.70 mm < Ø > 0.42 mm); MLDSF: medium-low date seed flour (0.42 mm < Ø > 0.21 mm); and LDSF: lowest date seed flour (Ø < 0.21 mm). Values with a different lowercase letter within the same column indicate significant differences (*p* < 0.05) according to Tukey’s multiple range test. Values followed with a different capital letter in the same row indicate significant differences (*p* < 0.05) according to Tukey’s multiple range test.

**Table 4 plants-13-00335-t004:** Effect of particle size on physicochemical properties of date seed flour cv. Medjool cultivated on the Elche palm grove.

	pH	Aw	Color Coordinates		
			L*	a*	b*
HDSF	5.55 ± 0.09 ^b^	0.523 ± 0.000 ^a^	51.98 ± 1.07 ^a^	8.53 ± 0.37 ^a^	11.84 ± 0.62 ^a^
MHDSF	4.86 ± 0.04 ^a^	0.534 ± 0.002 ^b^	53.48 ± 1.12 ^a^	9.09 ± 0.17 ^a^	13.09 ± 0.55 ^b^
MLDSF	4.88 ± 0.04 ^a^	0.544 ± 0.001 ^c^	53.27 ± 1.26 ^a^	9.81 ± 0.13 ^b^	13.36 ± 0.16 ^b^
LDSF	4.93 ± 0.03 ^a^	0.555 ± 0.003 ^d^	52.12 ± 0.23 ^a^	12.15 ± 0.07 ^c^	13.59 ± 0.08 ^b^

HDSF: highest date seed flour (1.16 mm < Ø > 0.70 mm); MHDSF: medium-high date seed flour (0.70 mm < Ø > 0.42 mm); MLDSF: medium-low date seed flour (0.42 mm < Ø > 0.21 mm); and LDSF: lowest date seed flour (Ø < 0.21 mm). Values with a different lowercase letter within the same column indicate significant differences (*p* < 0.05) according to Tukey’s multiple range test.

**Table 5 plants-13-00335-t005:** Effect of particle size on the polyphenolic profile of date seed flour cv. Medjool cultivated on the Elche palm grove.

	HDSF	MHDSF	MLDSF	LDSF
Protocatechuic acid	0.38 ± 0.02 ^dH^	0.48 ± 0.03 ^cF^	0.55 ± 0.04 ^bG^	0.65 ± 0.04 ^aH^
Catechin	10.57 ± 0.29 ^dA^	13.05 ± 0.32 ^cA^	16.93 ± 0.28 ^bA^	19.94 ± 0.15 ^aA^
Epicatechin	7.70 ± 0.15 ^dB^	9.55 ± 0.23 ^cB^	11.98 ± 0.19 ^bC^	15.67 ± 0.21 ^aB^
Epigallocatechin-3-gallate	6.27 ± 0.19 ^dC^	8.93 ± 0.20 ^cB^	13.19 ± 0.31 ^bB^	15.71 ± 0.22 ^aB^
Gallocatechin-3-gallate	0.54 ± 0.03 ^dG^	0.81 ± 0.05 ^cE^	1.31 ± 0.04 ^bE^	1.69 ± 0.07 ^aE^
caffeic acid	0.70 ± 0.06 ^cF^	0.72 ± 0.05 ^cEF^	0.92 ± 0.06 ^bF^	1.24 ± 0.08 ^aF^
5-*O*-Caffeoylshikimic acid	0.63 ± 0.05 ^cF^	0.67 ± 0.04 ^cF^	0.87 ± 0.03 ^bF^	1.23 ± 0.09 ^aF^
4-*O*-Caffeoylshikimic acid	0.36 ± 0.04 ^bH^	0.38 ± 0.04 ^bG^	0.37 ± 0.03 ^bH^	0.58 ± 0.05 ^aH^
3-*O*-Caffeoylshikimic acid	0.20 ± 0.02 ^aI^	0.19 ± 0.03 ^aH^	0.18 ± 0.02 ^aI^	0.18 ± 0.01 ^aI^
*p*-Coumaric	1.68 ± 0.12 ^cD^	2.07 ± 0.16 ^bC^	2.16 ± 0.13 ^bD^	2.77 ± 0.12 ^aC^
Quercetin 3-rutinoside	0.99 ± 0.08 ^cE^	1.15 ± 0.16 ^bD^	1.34 ± 0.12 ^bE^	2.01 ± 0.28 ^aD^
Quercetin 3-β-d-glucoside	0.51 ± 0.02 ^aG^	0.53 ± 0.02 ^aF^	0.54 ± 0.04 ^aG^	0.60 ± 0.04 ^aH^
Quercetin 3-rhamnoside	0.85 ± 0.03 ^aE^	0.83 ± 0.03 ^aE^	0.87 ± 0.02 ^aF^	0.84 ± 0.03 ^aG^

Values expressed as mg/100 g of date seed flour. HDSF: highest date seed flour (1.16 mm < Ø > 0.70 mm); MHDSF: medium-high date seed flour (0.70 mm < Ø > 0.42 mm); MLDSF: medium-low date seed flour (0.42 mm < Ø > 0.21 mm); and LDSF: lowest date seed flour (Ø < 0.21 mm). Values with a different lowercase letter within the same row indicate significant differences (*p* < 0.05) according to Tukey’s multiple range test. Values followed with a different capital letter in the same column indicate significant differences (*p* < 0.05) according to Tukey’s multiple range test.

## Data Availability

The data presented in this paper are available upon request from the corresponding author.

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
