# Peer review of "Date (Phoenix dactylifera L. cv. Medjool) Seed Flour, a Potential Ingredient for the Food Industry: Effect of Particle Size on Its Chemical, Technological, and Functional Properties"

_plants, 2024, doi:10.3390/plants13030335_

Round 1

Reviewer 1 Report

Comments and Suggestions for Authors

The Authors had a very practical task in their research and obtained very useful results. I think the manuscript is well written, but the introduction needs reorganization to make it more interesting for readers. In my opinion, the research was carried out correctly and the main goal of the research was achieved - the flour obtained from date seeds can be a valuable co-product for the food industry. Most sections were well-written, clear and complete. I have  some comments:

Introduction:

I think that the authors should rewrite the introduction, the description is too monotonous (line 52-line 75) and needs to be reorganized to make it more interesting for the reader. Please try to change it. Please also complete the information on the use of date flour or powder in the food industry.

Results

Please consider a different way of presenting data than in the form of tables. Please change it to make the data more accessible to readers.

Line 406: remove the date in brackets for the correct citation

Conclusions

Please expand further on the conclusions It would be helpful to mention potential future research directions or applications of this study's findings.

Ensure all references are cited properly in the text and listed correctly in the reference section. Review the article for grammar, punctuation, and clarity.

Author Response

First, at all, we would like to thank all the positive inputs and suggestions given by the reviewer, which will contribute to improve and enrich our manuscript. We have carefully undertaken the revision requests for your reconsideration. The answers are given just after the transcription of reviewers' comments and new information added to the manuscript was highlighted in red colour.

Revierwer #1

The Authors had a very practical task in their research and obtained very useful results. I think the manuscript is well written, but the introduction needs reorganization to make it more interesting for readers. In my opinion, the research was carried out correctly and the main goal of the research was achieved - the flour obtained from date seeds can be a valuable co-product for the food industry. Most sections were well-written, clear and complete. I have  some comments:

Introduction:

I think that the authors should rewrite the introduction, the description is too monotonous (line 52-line 75) and needs to be reorganized to make it more interesting for the reader. Please try to change it. Please also complete the information on the use of date flour or powder in the food industry.

Thank you for the suggestion. The introduction has been modified

Results

Please consider a different way of presenting data than in the form of tables. Please change it to make the data more accessible to readers.

Thank you for the suggestion. As the reviewer suggested, two tables (Techno-functional properties and antioxidant capacity) have transformed into figures

Line 406: remove the date in brackets for the correct citation

Many thanks for your comment. The correction has been done

Conclusions

Please expand further on the conclusions It would be helpful to mention potential future research directions or applications of this study's findings.

Many thanks for your comment. The suggestion has been added

Ensure all references are cited properly in the text and listed correctly in the reference section. Review the article for grammar, punctuation, and clarity.

Many thanks for your comment. The correction has been done, English language has been corrected

Reviewer 2 Report

Comments and Suggestions for Authors

The manuscript „plants-2814966” investigates the influence of particle size on the properties of flour obtained for date seeds. Chemical composition (moisture, protein, fat, ash, and total dietary fibre content), physicochemical properties (pH, colour and water activity), techno-functional properties (water-holding capacity, oil-holding capacity, and swelling capacity), polyphenol content, and antioxidant activity were determined for different particle sizes of date seed flour: highest (1160 mm < Ø > 0.70 mm), medium-high 0.70 mm < Ø > 0.42 mm), medium-low (0.42 mm < Ø > 0.21 mm), and lowest (Ø < 0.21 mm). Several properties, such as moisture and protein content and water and oil holding capacities, decrease with the reduction of particle sizes. Others, such as fat, minerals (Ca, Fe, Zn, and Mg) and fatty acid content, water activity, polyphenolic content, and antioxidant activity were significantly (p < 0.05) improved. The enhanced properties sustain the idea of using the flour from date seeds as an ingredient in food formulations to improve the sensory properties. Besides the added value of food products, using flour from date seeds helps reduce waste and promotes the circular economy in the food chain. However, no specific food formulation is mentioned in the manuscript.

The manuscript is structured and written well. The results are presented in seven tables and one figure, discussed and compared with data from the literature.

However, several shortcomings were identified and presented below.

Shortcomings of the manuscript

I. Keywords

„Coproducts” is not a very clever choice as a keyword.

The list of keywords could be improved by adding other specific keywords, such as „date seed flour”, „particle size”, or „functional properties”.

II. Section number

L311 Correct the number of subsection 3.6 with 2.5.

III. Grammar errors – are presented below in the section „Comments on the quality of English language.”

Comments on the Quality of English Language

The manuscript is written well using a clear English language. However, several minor language errors have been identified. A few examples are presented below.

Inconsistent hyphenation: The authors have written the same word with and without a hyphen: „co-product” (L28, 48, 49, and 73) and „coproduct” (L31, 193, and 217). Both ways are acceptable, but it is best to be consistent.

Correct „low cas” (L153, 177, 212) and „low case” (L292) with „lower case” or „lowercase.”

Correct „indicates” with „indicate” (the noun „values” is plural). (L113, 153, 155, 177, 179, 212, 246, 292, 294, 324)

Missing commas (e.g., before „which”, „but”, „as well as”, etc.) has to be corrected.

L19 Consider adding the indefinite article „a” before „high.”

L19 Consider changing the form of the verb „decrease” to „decreased.”

L20 The verb „were” does not seem to agree with the subject „content”. Consider changing the verb form „were” to „was.”

L22 Consider changing the form of the verb „tend” to „tended.”

L23 Consider changing the adjective „highest” to „largest”, which is more appropriate with the noun „sizes.”

L24-25 Consider moving the introductory phrase „In all samples” after „compounds” and correct „cathechin” with „catechin”: „The main polyphenolic compounds in all samples were catechin, epicatechin and Epigallocatechin-3-gallate.”

L30 Insert a comma after „generated.”

L34 Consider removing the definite article „The” before „Eco-efficiency.”

L35 Consider removing the indefinite article „a” before „sustainable.”

L38 Consider removing „production” before „process” to avoid repetition and clarify the sentence.

L42 Consider adding a comma after „foods” (before „as well as”) and removing „to” before „preserve.”

L44 Consider adding a comma after „community” (before „which”)

L46-49 Consider rephrasing the sentence. Suggestions:

a) Insert commas after „Globalization” and „trends”: „Globalization, along with the emerging consumption trends, has caused the food industry to increase the use of raw materials or primary products, leading to an increase in co-products and creating problems with their management and utilization.”

b) Replace „along with the” with „and” and the verb „has” with „have”: „Globalization and emerging consumption trends have caused the food industry to increase the use of raw materials or primary products, leading to an increase in co-products and creating problems with their management and utilization.

L52 Consider removing the comma after „date” and adding a comma before „originating.”

L56 Consider replacing „sustained” with „supported.”

L58 Consider rewording the sentence: use the plural „dates”, remove the indefinite article „a”, and use the plural „fruits.”

L64 Remove the hyphen in „high-water.”

L66-70 Consider rewording the sentence: replace „This” with „The”, use the plural „dates”, add a comma after „season”, and replace „ as a result of not meet” with „due to not meeting.”

L70 Consider adding commas before and after „therefore.” The phrase „period of time” may be redundant. Consider removing „of time.”

L73 Consider removing the definite article „the” before „date seeds.”

L74 Consider removing „they” or splitting the sentence.

L75 Consider removing „food” before „manufacturing” to avoid the repetition.

L77 Consider replacing the verb form „make” with „makes.”

L78-80 The sentence may sound redundant. Consider removing „of great importance.”

L80-82 Consider rephrasing the sentence. Insert „composition of” before „oil” and remove „its composition.”

L82-83 Consider replacing „oleic acid and linoleic acid” with „oleic and linoleic acids” to avoid the repetition.

L85 Consider adding a comma after „phenols.”

L90-92 Consider rephrasing the sentence. Replace „conducted” with „conducting”, add a comma after „limited”, and replace „and especially in the case of” with „especially regarding.”

L98 Consider adding the definite article „the” before „chemical” and changing the form of the verb „was showed” to „is shown” (the past tense is not proper for this sentence and the past participle of the verb „to show” is „shown”).

L101 Split the sentence, replacing the comma after „incorporated” with a dot and replacing „being” with „It is.”

L102 Consider adding a comma after „analyzed.”

L105 Consider adding a comma before „as stated.”

L112-114 Consider moving „According to Tukey’s Multiple Range Test” at the beginning of the sentence (before „values with”). Also, correct the form of the verb „indicates”, which is singular, with „indicate.”

L121 Consider adding a comma after „(Table 1).”

L126 Consider changing the wording „In relation to” to „Concerning” and adding a comma after „(Table 1).”

L128 Consider adding a comma before „while” and „of” between „size” and „date.”

L141 Consider changing the form of the verb „showed” to „shows.”

L142-146 Rewrite the sentences for clarity.

L156 Consider adding commas after „analyzed” and „values.”

L157 Consider removing „the” before „zinc” and adding a comma before „which.”

L163 Consider changing „cooper” to „copper.”

L181 Consider adding a comma before „increasing.”

L187 Consider changing the form of the verb „have” to „has.”

L189, 190 Consider changing „its” to „their.”

L193 Consider changing the form of the verb „play” to „plays.”

L195 Consider replacing „interesting” with a synonym, e.g., „attractive” and „with the aim of” with „to.”

L196 Consider adding a comma before „such” and replacing „of” with „to.”

L197 Consider replacing „resulted to be” with „was.”

L199 Consider adding a comma before „although.”

L201 Consider changing the form of the verb „showed” to „shows.”

L203 Consider adding a comma after „flours.”

L214 Consider correcting „decreased” to „decrease” (it is a noun).

L215 Consider adding a comma after „increase” and mobbing „being” after „LDSF.”

L216 Consider changing the form of the verb „has” to „have” (the noun „values” is plural).

L235 Consider replacing „for” with „by.”

L238 Consider replacing „in” with „on.”

L269 Consider changing the form of the verb „showed” to „shows.”

L274 Consider changing the form of the verb „had” to „has.”

L281-282 Consider replacing „Caffeic acida and p-coumaric acid” with „Caffeic and p-coumaric acids.”

L299 Consider correcting „an dextracted” with „and extracted.”

L313 Consider replacing „of” with „in” before „the results.”

L328 Consider replacing „be due to the fact that” with „because.”

L330 Consider changing the form of the verb „were” to „was.”

L336 Consider correcting „Similarlty” with Similarly.”

L339 Consider adding a comma after „studies.”

L339-340 Consider removing one of „values.”

L354 Consider changing the form of the verb „showed” to „shows.”

L378 Consider adding „of” after „angle.”

L384 Consider removing „s” from „particles” – the plural is provided by „sizes.”

L394 Consider replacing „during” with „for.”

L401 Consider changing the form of the verb „describe” to „described.”

L407 Add a period at the end of the sentence.

Author Response

First, at all, we would like to thank all the positive inputs and suggestions given by the reviewer, which will contribute to improve and enrich our manuscript. We have carefully undertaken the revision requests for your reconsideration. The answers are given just after the transcription of reviewers' comments and new information added to the manuscript was highlighted in red colour.

Revierwer #2

The manuscript „plants-2814966” investigates the influence of particle size on the properties of flour obtained for date seeds. Chemical composition (moisture, protein, fat, ash, and total dietary fibre content), physicochemical properties (pH, colour and water activity), techno-functional properties (water-holding capacity, oil-holding capacity, and swelling capacity), polyphenol content, and antioxidant activity were determined for different particle sizes of date seed flour: highest (1160 mm < Ø > 0.70 mm), medium-high 0.70 mm < Ø > 0.42 mm), medium-low (0.42 mm < Ø > 0.21 mm), and lowest (Ø < 0.21 mm). Several properties, such as moisture and protein content and water and oil holding capacities, decrease with the reduction of particle sizes. Others, such as fat, minerals (Ca, Fe, Zn, and Mg) and fatty acid content, water activity, polyphenolic content, and antioxidant activity were significantly (p < 0.05) improved. The enhanced properties sustain the idea of using the flour from date seeds as an ingredient in food formulations to improve the sensory properties. Besides the added value of food products, using flour from date seeds helps reduce waste and promotes the circular economy in the food chain. However, no specific food formulation is mentioned in the manuscript.  The manuscript is structured and written well. The results are presented in seven tables and one figure, discussed and compared with data from the literature.

Thanks for the suggestion

However, several shortcomings were identified and presented below.

Shortcomings of the manuscript

  1. Keywords

„Coproducts” is not a very clever choice as a keyword.

Thanks for the suggestion, Coproducts has been removed

The list of keywords could be improved by adding other specific keywords, such as „date seed flour”, „particle size”, or „functional properties”.

Thanks for the suggestion. These keywords have been added to the text

  1. Section number

L311 Correct the number of subsection 3.6 with 2.5.

Thanks for the correction

III. Grammar errors – are presented below in the section „Comments on the quality of English language.” Comments on the Quality of English Language

The manuscript is written well using a clear English language. However, several minor language errors have been identified. A few examples are presented below.

The English language has been corrected

Inconsistent hyphenation: The authors have written the same word with and without a hyphen: „co-product” (L28, 48, 49, and 73) and „coproduct” (L31, 193, and 217). Both ways are acceptable, but it is best to be consistent.

The correction has been done

Correct „low cas” (L153, 177, 212) and „low case” (L292) with „lower case” or „lowercase.”

Many thanks for your comment. The correction has been done

Correct „indicates” with „indicate” (the noun „values” is plural). (L113, 153, 155, 177, 179, 212, 246, 292, 294, 324)

Many thanks for your comment. The correction has been done

Missing commas (e.g., before „which”, „but”, „as well as”, etc.) has to be corrected.

Many thanks for your comment. The correction has been done

L19 Consider adding the indefinite article „a” before „high.”

Many thanks for your comment. The correction has been done

L19 Consider changing the form of the verb „decrease” to „decreased.”

Many thanks for your comment. The correction has been done

L20 The verb „were” does not seem to agree with the subject „content”. Consider changing the verb form „were” to „was.”

Many thanks for your comment. The correction has been done

L22 Consider changing the form of the verb „tend” to „tended.”

Many thanks for your comment. The correction has been done

L23 Consider changing the adjective „highest” to „largest”, which is more appropriate with the noun „sizes.”

Many thanks for your comment. The correction has been done

L24-25 Consider moving the introductory phrase „In all samples” after „compounds” and correct „cathechin” with „catechin”: „The main polyphenolic compounds in all samples were catechin, epicatechin and Epigallocatechin-3-gallate.”

Many thanks for your comment. The correction has been done

L30 Insert a comma after „generated.”

Many thanks for your comment. The correction has been done

L34 Consider removing the definite article „The” before „Eco-efficiency.”

Many thanks for your comment. The correction has been done

L35 Consider removing the indefinite article „a” before „sustainable.”

Many thanks for your comment. The correction has been done

L38 Consider removing „production” before „process” to avoid repetition and clarify the sentence.

Many thanks for your comment. The correction has been done

L42 Consider adding a comma after „foods” (before „as well as”) and removing „to” before „preserve.”

Many thanks for your comment. The correction has been done

L44 Consider adding a comma after „community” (before „which”)

Many thanks for your comment. The correction has been done

L46-49 Consider rephrasing the sentence. Suggestions: Insert commas after „Globalization” and „trends”: „Globalization, along with the emerging consumption trends, has caused the food industry to increase the use of raw materials or primary products, leading to an increase in co-products and creating problems with their management and utilization.”

Many thanks for your comment. The correction has been done

L52 Consider removing the comma after „date” and adding a comma before „originating.”

Many thanks for your comment. The correction has been done

L56 Consider replacing „sustained” with „supported.”

Many thanks for your comment. The correction has been done

L58 Consider rewording the sentence: use the plural „dates”, remove the indefinite article „a”, and use the plural „fruits.”

Many thanks for your comment. The correction has been done

L64 Remove the hyphen in „high-water.”

Many thanks for your comment. The correction has been done

L66-70 Consider rewording the sentence: replace „This” with „The”, use the plural „dates”, add a comma after „season”, and replace „ as a result of not meet” with „due to not meeting.”

Many thanks for your comment. The correction has been done

L70 Consider adding commas before and after „therefore.” The phrase „period of time” may be redundant. Consider removing „of time.”

Many thanks for your comment. The correction has been done

L73 Consider removing the definite article „the” before „date seeds.”

Many thanks for your comment. The correction has been done

L74 Consider removing „they” or splitting the sentence.

Many thanks for your comment. The correction has been done

L75 Consider removing „food” before „manufacturing” to avoid the repetition.

Many thanks for your comment. The correction has been done

L77 Consider replacing the verb form „make” with „makes.”

Many thanks for your comment. The correction has been done

L78-80 The sentence may sound redundant. Consider removing „of great importance.”

Many thanks for your comment. The correction has been done

L80-82 Consider rephrasing the sentence. Insert „composition of” before „oil” and remove „its composition.”

Many thanks for your comment. The correction has been done

L82-83 Consider replacing „oleic acid and linoleic acid” with „oleic and linoleic acids” to avoid the repetition.

Many thanks for your comment. The correction has been done

L85 Consider adding a comma after „phenols.”

Many thanks for your comment. The correction has been done

L90-92 Consider rephrasing the sentence. Replace „conducted” with „conducting”, add a comma after „limited”, and replace „and especially in the case of” with „especially regarding.”

Many thanks for your comment. The correction has been done

L98 Consider adding the definite article „the” before „chemical” and changing the form of the verb „was showed” to „is shown” (the past tense is not proper for this sentence and the past participle of the verb „to show” is „shown”).

Many thanks for your comment. The correction has been done

L101 Split the sentence, replacing the comma after „incorporated” with a dot and replacing „being” with „It is.”

Many thanks for your comment. The correction has been done

L102 Consider adding a comma after „analyzed.”

Many thanks for your comment. The correction has been done

L105 Consider adding a comma before „as stated.”

Many thanks for your comment. The correction has been done

L112-114 Consider moving „According to Tukey’s Multiple Range Test” at the beginning of the sentence (before „values with”). Also, correct the form of the verb „indicates”, which is singular, with „indicate.”

Many thanks for your comment. The correction has been done

L121 Consider adding a comma after „(Table 1).”

Many thanks for your comment. The correction has been done

L126 Consider changing the wording „In relation to” to „Concerning” and adding a comma after „(Table 1).”

Many thanks for your comment. The correction has been done

L128 Consider adding a comma before „while” and „of” between „size” and „date.”

Many thanks for your comment. The correction has been done

L141 Consider changing the form of the verb „showed” to „shows.”

Many thanks for your comment. The correction has been done

L142-146 Rewrite the sentences for clarity.

Many thanks for your comment. The correction has been done

L156 Consider adding commas after „analyzed” and „values.”

Many thanks for your comment. The correction has been done

L157 Consider removing „the” before „zinc” and adding a comma before „which.”

Many thanks for your comment. The correction has been done

L163 Consider changing „cooper” to „copper.”

Many thanks for your comment. The correction has been done

L181 Consider adding a comma before „increasing.”

Many thanks for your comment. The correction has been done

L187 Consider changing the form of the verb „have” to „has.”

Many thanks for your comment. The correction has been done

L189, 190 Consider changing „its” to „their.”

Many thanks for your comment. The correction has been done

L193 Consider changing the form of the verb „play” to „plays.”

Many thanks for your comment. The correction has been done

L195 Consider replacing „interesting” with a synonym, e.g., „attractive” and „with the aim of” with „to.”

Many thanks for your comment. The correction has been done

L196 Consider adding a comma before „such” and replacing „of” with „to.”

Many thanks for your comment. The correction has been done

L197 Consider replacing „resulted to be” with „was.”

Many thanks for your comment. The correction has been done

L199 Consider adding a comma before „although.”

Many thanks for your comment. The correction has been done

L201 Consider changing the form of the verb „showed” to „shows.”

Many thanks for your comment. The correction has been done

L203 Consider adding a comma after „flours.”

Many thanks for your comment. The correction has been done

L214 Consider correcting „decreased” to „decrease” (it is a noun).

Many thanks for your comment. The correction has been done

L215 Consider adding a comma after „increase” and mobbing „being” after „LDSF.”

Many thanks for your comment. The correction has been done

L216 Consider changing the form of the verb „has” to „have” (the noun „values” is plural).

Many thanks for your comment. The correction has been done

L235 Consider replacing „for” with „by.”

Many thanks for your comment. The correction has been done

L238 Consider replacing „in” with „on.”

Many thanks for your comment. The correction has been done

L269 Consider changing the form of the verb „showed” to „shows.”

Many thanks for your comment. The correction has been done

L274 Consider changing the form of the verb „had” to „has.”

Many thanks for your comment. The correction has been done

L281-282 Consider replacing „Caffeic acida and p-coumaric acid” with „Caffeic and p-coumaric acids.”

Many thanks for your comment. The correction has been done

L299 Consider correcting „an dextracted” with „and extracted.”

Many thanks for your comment. The correction has been done

L313 Consider replacing „of” with „in” before „the results.”

Many thanks for your comment. The correction has been done

L328 Consider replacing „be due to the fact that” with „because.”

Thanks for the suggestion, we prefer maintain due to

L330 Consider changing the form of the verb „were” to „was.”

Many thanks for your comment. The correction has been done

L336 Consider correcting „Similarlty” with Similarly.”

Many thanks for your comment. The correction has been done

L339 Consider adding a comma after „studies.”

Many thanks for your comment. The correction has been done

L339-340 Consider removing one of „values.”

Many thanks for your comment. The correction has been done

L354 Consider changing the form of the verb „showed” to „shows.”

Many thanks for your comment. The correction has been done

L378 Consider adding „of” after „angle.”

Many thanks for your comment. The correction has been done

L384 Consider removing „s” from „particles” – the plural is provided by „sizes.”

Many thanks for your comment. The correction has been done

L394 Consider replacing „during” with „for.”

Many thanks for your comment. The correction has been done

L401 Consider changing the form of the verb „describe” to „described.”

Many thanks for your comment. The correction has been done

 L407 Add a period at the end of the sentence.

Many thanks for your comment. The correction has been done

Round 2

Reviewer 2 Report

Comments and Suggestions for Authors

The authors revised the manuscript „plants-2814966” following the recommendations and observations of the reviewers. Thus, they analyzed and applied the suggestions for English grammar errors and improved the keywords and the introduction. The corrections are coloured red and are easy to notice. Therefore, the improved version is qualified for publication.